# Effect of Yb^3+^ on the Structural and Visible to Near-Infrared Wavelength Photoluminescence Properties in Sm^3+^-Yb^3+^-Codoped Barium Fluorotellurite Glasses

**DOI:** 10.3390/ma15093314

**Published:** 2022-05-05

**Authors:** Eric Kumi-Barimah, Yan Chen, Rebekah Tenwick, Mohanad Al-Murish, Geeta Sherma, Animesh Jha

**Affiliations:** School of Chemical and Process Engineering, University of Leeds, Clarendon Road, Leeds LS2 9JT, UK; pm20yc@leeds.ac.uk (Y.C.); cm15rjt@leeds.ac.uk (R.T.); fy13maam@leeds.ac.uk (M.A.-M.); g.sharma@leeds.ac.uk (G.S.); a.jha@leeds.ac.uk (A.J.)

**Keywords:** rare-earth ions, barium fluorotellurite glasses, hydroxyl (-OH), bonding parameters, photoluminescence and lifetime

## Abstract

We report on the Sm^3+^ and Sm^3+^:Yb^3+^-doped barium fluorotellurite glasses prepared using the conventional melting and quenching method. The spectroscopic characterisations were investigated with Raman and FTIR to evaluate the glasses’ structural and hydroxyl (-OH) content. The Raman analysis revealed a structural modification in the glass network upon adding and increasing the Yb^3+^ concentration from a TeO_3_ trigonal pyramid to a TeO_4_ trigonal bi-pyramid polyhedral. At the same time, the FTIR measurements showed the existence of -OH groups in the glass. Thus, under the current experimental conditions and nominal composition, the -OH group contents are too large to enable an effective removal. The near-infrared region of the absorption spectra is employed to determine the nephelauxetic ratio and bonding parameters. The average nephelauxetic ratio decreases, and the bonding parameter increases with the increasing Yb^3+^ content in the glasses. A room temperature visible and near-infrared photoluminescence ranging from 500 to 1500 nm in wavelength and decay properties were investigated for glasses doped with Sm^3+^ and Sm^3+^-Yb^3+^ by exciting them with 450 and 980 nm laser sources. Exciting the Sm^3+^- and Sm^3+^-Yb^3+^-doped glasses by 450 nm excitation reveals a new series of photoluminescence emissions at 1200, 1293, and 1418 nm, corresponding to the ^6^F_11/2_ state to the ^6^H*_J_* (*J* = 7/2, 9/2, 11/2) transitions. Under the 976 nm laser excitation, a broad photoluminescence emission from 980 to 1200 nm was detected. A decay lifetime decreased from ~244 to ~43 μs by increasing the Yb^3+^ content, ascribing to concentration quenching and the OH content.

## 1. Introduction

During the last decades, various rare-earth (RE^3+^) ions doped in different host-glass matrices have been investigated for applications such as laser sources for optical telecommunications, optical amplifiers, multicolour displays, and medical diagnostics (sensing and imaging in the anatomical environment) [1,2,3,4]. Tellurium oxide (TeO_2_)-based glasses have numerous advantages over the borate, phosphate, and silicate host glasses [1,2,5]. This is due to their unique high linear and nonlinear refractive indices; outstanding characteristics, including high density, thermal stability, crystallisation, chemical durability, and low melting temperature; and high optical transparency in the visible to IR region up to 0.35–5 μm [5]. In addition, TeO_2_-based glasses have a high stimulated emission cross-section, a low-phonon-energy range from 560 to 750 cm^−1^ depending on the lattice modifier, and a high RE^3+^-ion concentration solubility [6,7,8,9]. However, a pure TeO_2_ has a low glass-forming ability due to its high quenching rate [10,11,12,13]. Consequently, this requires lattice modifiers such as alkali oxides, alkaline earth oxides, and transition metal oxides and halides to vitrify with a moderate quenching rate [13]. These lattice modifiers enable the TeO_4_ and TeO_3_ structural units to break down and decrease the average Te-O coordination [14], thus enhancing the glass-forming ability. For instance, adding a barium fluoride (BaF_2_) modifier to a TeO_2_-based glass network tends to decrease the phonon energy remarkably as compared to the phonon energy of the TeO_2_-based (Te-O~750 cm^−1^) [15,16]. Thus, reducing the nonradiative and multiphonon relaxation mechanism promotes optical transitions in RE^3+^ ions.

RE^3+^ ions such as trivalent samarium (Sm^3+^) are among the most studied active ions with low- and high-phonon-energy host materials owing to their adjacent closely lying energy-level structures, the enormous energy difference between the metastable level ^4^G_5/2_, and lower-lying levels such as ^6^F*_J_* (*J* = 1/2, 3/2, 5/2,7/2, 9/2, and 11/2) and ^6^H*_J_* (*J* = 5/2,7/2, 9/2, and 11/2) [16,17]. In addition, Sm^3+^ has more NIR absorption and photoluminescence emission transitions than erbium (Er^3+^), neodymium (Nd^3+^), and ytterbium (Yb^3+^). Hence, exploiting absorption and photoluminescence emission properties within the 700–1300 nm optical window [18] is attractive for monitoring soft tissue molecular signatures because of the low absorption and scattering in this spectral range. In addition, it has a vast potential for developing NIR lasers, optical amplifiers, temperature sensors, and biomedical devices.

Recently, numerous host materials doped with a Sm^3+^ ion have been explored because of their interesting optical properties and broad photoluminescence band from ^4^G_5/2_ to ^6^H_J_ and ^6^F_J_ transitions [19,20,21,22,23,24,25,26,27,28,29]. Here are some current studies of Sm^3+^-doped glasses with the focussed-on fluorescence emission properties from the orange to red wavelength region. Mawlud et al. [5] had reported optical spectroscopic properties on various concentrations of Sm^3+^ ion-doped TeO_2_-Na_2_O glasses. They demonstrated an increasing photoluminescence emission intensity under a 404 nm excitation source at the following transitions, ^4^G_5/2_ → ^6^H_5/2_ (561 nm), ^4^G_5/2_ → ^6^H_7/2_ (598 nm), ^4^G_5/2_ → ^6^H_9/2_ (645), and ^4^G_5/2_ → ^6^H_11/2_ (704 nm), upon increasing the Sm^3+^ ion concentration. Similarly, Hussain et al. [23] prepared various Sm^3+^ ion-doped phosphate glasses to analyse their optical absorption, radiative transition probability, and photoluminescence properties. The four main visible photoluminescence emission transitions from ^4^G_5/2_ to ^6^H_J_ were observed under an excitation of 400 nm. Even though the photoluminescence properties of Sm^3+^-doped glasses have been studied significantly between the 400 and 700 nm spectral region, there is limited research on the NIR transitions ranging from 800 to 1500 nm for comparative studies. Recently, Herrara et al. [24] investigated the structural and photoluminescence properties of Sm^3+^-doped heavy-metal oxide glass comprised of B_2_O_3_-PbO-Bi_2_O_3_-GeO_2_. The photoluminescence spectral of the glass revealed strong emission peaks of the Sm^3+^ ion at 563, 601, 648, and 711 nm, respectively. Additionally, NIR photoluminescence bands centred at 912, 954, 1036, and 1182 nm were reported, which matched with transitions from ^4^G_5/2_ → ^6^F*_J_* (*J* = 3/2, 5/2, 7/2, 9/2). Likewise, visible and NIR photoluminescence emission peaks from tellurite zinc oxide glass doped with Sm^3+^ ions under an excitation wavelength of 403 nm had also been reported elsewhere [25].

The current approach to increasing the narrow absorption cross-section of the Sm^3+^ ions energy level in the NIR region from 850 to 1000 nm is codoping with Yb^3+^ to develop a long-wavelength broadband NIR (850–1150 nm) glass with enhanced optical properties through the resonant energy transfer process. Yb^3+^ is a well-known efficient sensitiser and activator with a broad absorption cross-section ranging from 950 to 1060 nm in wavelength, corresponding to the Yb^3+^: ^6^F_7/2_ → ^6^F_5/2_ transition. Therefore, codoping the Sm^3+^: Yb^3+^ with glass could broaden the absorption cross-section from 800 to 1150 nm by overlapping the Sm^3+^ (^4^G_5/2_ → ^6^F_3/2_, ^4^G_5/2_ → ^6^F_5/2_, ^4^G_5/2_ → ^6^F_7/2_) and Yb^3+^ (^6^F_7/2_ → ^6^F_5/2_) transitions. This approach would increase the pump efficiency to enable the realisation of population inversion for optical amplification via energy transfer processes between Sm^3+^ and Yb^3+^, curtailing resonant nonradiative transfer or nonresonant phonon-assisted nonradiative transfer [26,27,28].

In this article, we have fabricated Sm^3+^- and Yb^3+^:Sm^3+^-doped barium fluorotellurite zinc glasses by varying the Yb^3+^ concentrations. We performed a comprehensive analysis to investigate the influence of the Yb^3+^ content on the barium fluorotellurite zinc glasses’ structure and -OH content via Raman spectroscopy and Fourier transform infrared (FTIR) transmittance and absorption spectra. The Yb^3+^ concentration effect on the glass refractive index and nephelauxetic ratios are also determined. Furthermore, the visible and NIR photoluminescence emission and decay lifetimes are examined to understand the energy transfer process between the Yb^3+^ and Sm^3+^ ions by increasing the Yb^3+^ concentration under 450 and 980 nm pumping schemes.

## 2. Materials and Method

Barium fluorotellurite glasses doped/codoped with Sm^3+^ and Sm^3+^: Yb^3+^ ions with molar composition of (97.5 − x)TeO_2_-10ZnO-10BaF_2_-0.5Sm_2_O_3_-(x = 0–1.0Yb_2_O_3_) were fabricated using conventional melting and quenching technique. The starting materials comprise high-purity TeO_2_ (≥99.99%), ZnO (99.99%), BaF_2_ (99.99%), Yb_2_O_3_ (99.99%), and Sm_2_O_3_ (99.99%), which were purchased from Alfa Aesar. A 30 g batch of Sm^3+^- and Sm^3+^: Yb^3+^-doped barium fluorotellurite glasses were synthesised by weighing the appropriate amount of each powder raw material. A homogenous mixture of the powder samples was obtained using a mortar and a pestle to mix thoroughly for about 20 min. The mixture was transferred into a gold crucible and melted at 750–800 °C for about 3 h by employing an electrical furnace (Elite Thermal Systems Limited, Market Harborough, UK). During the glass melting, the furnace atmosphere was maintained under a dry oxygen atmosphere by purging a high-purity oxygen gas to control moisture and OH-ion ingress in the glass melt. The homogeneous molten glass was poured into a preheated brass mould at 300 °C for 3 h and then annealed at 300 °C for 4 h to remove thermal stress.

Finally, the annealing furnace was allowed to cool down to room temperature overnight and polished for optical measurements. A prism coupler (Metricon model 2010/M) and Thermo Pycnomatic ATC Helium Pycnometer were also utilised to measure the glass’s linear refractive index and density. The molar composition, density, and refractive index of the achieved glasses are depicted in Table 1. Raman spectroscopy measurements were performed utilising a Renishaw via Raman spectrometer with a green Ar^+^ laser (λ = 514.5 nm) excitation source (Renishaw, Wotton-under-Edge, UK) to determine the molecular structure of as-prepared glasses within the spectral range of 100–1000 cm^−1^. Similar, infrared absorbance spectroscopy was conducted to investigate the effect of the single-doped local structures compared to codoped glasses. The infrared absorbance spectra were collected in a Vertex 70 FITR spectrometer (Brucker, Coventry, UK) over 4 cm^−1^ resolution between 12,000 and 400 cm^−1^. The visible and NIR photoluminescence properties were investigated at room temperature with an FS920 spectrometer (Edinburgh Instruments, Livingston, UK) by exciting with 450 and 980 nm laser sources in the wavelength range of 500–750 and 850–1500 nm. The visible and NIR photoluminescence emission measurements were performed with a 495 nm short-pass filter cut-off wavelength.

## 3. Results and Discussion

### 3.1. Physical and Structural Properties—Density, Refractive Index, Raman and FTIR Spectroscopies

Table 1 shows the measured densities and the refractive index as a function of the Y^3+^ ion concentration. The average density values were determined using the Thermo Pycnomatic ATC Helium Pycnometer instrument range from 5.632 to 5.618 g/cm^3^. The refractive index increases slightly as the content of the Yb^3+^ increases, which could be attributed to its dependence on the polarisability of the ions or electron density [29] in the glass

Raman spectroscopy is a well-known technique for determining the host materials’ vibrational, structural, and functional groups and the phonon energy for the optical applications. Hence, the Raman spectra of the SBT glass series with different Yb^3+^ concentrations were monitored within a wavenumber spectral range of 0–1000 cm^−1^, as depicted in Figure 1. The Boson peak of the tellurite glass due to the disordered structure of the amorphous state occurring between 60 and 100 cm^−1^ is observed in these glass samples.

It can be noticed that the Raman spectra of all the glasses prepared exhibit a broad band between 0 and 200 cm^−1^ and a strong peak centred at ~65 cm^−1^, which confirms the presence of the Boson peak in the SBT series’ glassy structure. This apparent characteristic peak can be attributed to acoustic phonon and rotational molecular modes [29]. The Raman spectra of all the glass samples prepared were characterised by deconvoluting three prominent broad bands ranging from 220 to 370, 370 to 546, and 546 to 900 cm^−1^, as shown in Figure 1a–d, using origin software and fitted with Gaussian sub-band curves. The weak Raman intensity observed at 295 cm^−1^ can be attributed to the Ag and Fg Raman active modes of Sm-O [30,31]. The vibration band that peaked at 450 cm^−1^ is assigned to the bending and stretching vibrations of the Te-O-Te or O-Te-O linkage bonds, increasing the amplitude by increasing the Yb^3+^ content. On the other hand, enhancing this prominent peak could be attributed to the formation of Te-O-Ba/Yb sharing a vertex with TeO_4_, TeO_3+δ_, and TeO_3_ [32,33,34]. In addition, the vibration bands centred at 664 and 734 cm^−1^ show a vigorous Raman intensity and are ascribed to the structural fragment of the TeO_4_ trigonal bi-pyramid polyhedral (tbps), TeO_3_ trigonal pyramid (tp), and intermediate TeO_3+δ_ polyhedron units, respectively [2,31,33,34]. The TeO_4_ (tbps) vibration peak is accredited to the asymmetric stretching vibrations of the Te-O-Te bridges which connect the TeO_3_ tbps at the vertices, whereas the TeO_3_ (tp) vibration band represents symmetric and asymmetric stretching modes due to the formation of nonbridge oxygen. It can be noticed that the relative intensity of a Sm^3+^-doped barium fluorotellurite (sample SBT1) Raman peak centred at 734 cm^−1^ (TeO_3_) is more intense as compared to the vibration band that occurred at 664 cm^−1^ (TeO_4_). The vibrational band centred at 781 cm^−1^ is ascribed to the stretching mode of the nonbridging oxygen TeO_3+1_ unit. Nevertheless, upon doping and increasing the Yb^3+^ concentration, the amplitude of the vibration band at 664 cm^−1^ progressively increases and predominates over the 734 cm^−1^ peak, as shown in Figure 2a. This demonstrates that the TeO_3_ (tps) unit has undergone a structural transformation by reducing the network connectivity to the TeO_4_ unit in the glass network. Figure 2b illustrates the intensity ratio of the bridge oxygen to nonbridge oxygen atoms (I664I735) as a function of the Yb_2_O_3_ concentration (mol%). It was observed that an increase in the Yb_2_O_3_ concentration shows a slight enhancement in the TeO_4_ (tbps) vibration band amplitude without any shift in the wavenumber. According to Maaoui et al. [35], such behaviour could be due to F^−^ (149 pm) substituting O^2−^ (152 pm) because of the similarities in the ionic radius. For instance, the network form can be broken down by replacing two F- with one oxygen atom or interchanging F- with Ba in the TeO3 (tps) to become Te(O, F)3 someplace within the glass network.

Figure 3a illustrates a mid-infrared transmittance spectra of samples SBT1, SBT2, SBT3, and SBT4 with corresponding thicknesses of 4.45, 4.58, 4.56, and 4.48 mm recorded using a Brucker Vertex 70 FTIR spectrometer. Sample SBT1 at 4.45 mm thick exhibits a high average transmittance of about 52% within the spectral range of 2.5 to 3.0 μm and longer mid-infrared cut-off edges near 6.25 μm. The transmission of sample SBT1 revealed two broad bands dipping around 3.3 μm (3000 cm^−1^) and 4.38 μm (2304 cm^−1^), which are assigned to the -OH groups connected in the glass network [36,37,38]. The absorption band that occurred at 3000 cm^−1^ is attributed to the free -OH groups coupled to cations in the glass network, with a typical example being tellurium ion (Te-OH) and weakly bonded with hydrogen (H-OH) [36]. The absorption peak centred at 2304 cm^−1^ is due to the -OH absorption strongly connected with hydrogen (H-OH) and multiphonon absorption. However, no noticeable changes are observed upon doping with various concentrations of Yb^3+^ with sample STBT1; nevertheless, transmission decreases with decreasing Yb^3+^ content in the glass, as shown in Figure 3a.

The absorption coefficient of -OH, $\alpha_{\mathrm{OH}}\left({\mathrm{cm}}^{-1}\right)$ in the fluorotellurite glass was estimated by employing the Beer–Lambert law equation, which is expressed as [36]:(1)αOH=ln(T(%)100)L=AL

Meanwhile, the corresponding concentration $N_{{\mathrm{OH}}^{-}}\left(\mathrm{ions}/{\mathrm{cm}}^{-3}\right)$ is obtained using Equation (2) [37]
(2)$$N_{{\mathrm{OH}}^{-}}=\frac{N_{AVO}}{\varepsilon }\alpha_{\mathrm{OH}}$$
where *T*(%) is the maximum transmittance in percentage at 3000 and 2304 cm^−1^, L is the glass thickness (cm), A is the absorbance, ε is the -OH group molar absorptivity in the tellurite glass (4.91 × 10^4^ cm^2^mol^−1^) [38], and *N_AVO_* represents Avogadro’s constant (6.02 × 10^23^ mol^−1^). 

The absorption coefficient spectra were determined using the transmittance spectra and Equation (1), as depicted in Figure 3b. The maximum absorption peaks centred at 3.30 and 4.38 μm in Figure 3b were employed to estimate the -OH concentration in all the glasses prepared, which is summarised in Table 2. It is observed that the $\alpha_{\mathrm{OH}}$(cm^−1^) and $N_{{\mathrm{OH}}^{-}}\left(\mathrm{ions}/{\mathrm{cm}}^{-3}\right)$ increase with increasing Yb^3+^ content in the glass series and these values correlate with oxyfluoride tellurite glasses reported elsewhere, respectively [35]. Therefore, high hydroxide absorption in the glasses can be reduced by either increasing the BaF_2_ content or substituting a portion of the ZnO with ZnF_2_.

### 3.2. Absorption Spectra, Bonding between Sm^3+^ and Surrounding Oxygen Atoms

Figure 4a represents the absorption coefficient spectra of the SBT glass series measured at room temperature using a FITIR spectrometer in the wavenumber region of 4000 to 12,000 cm^−1^. The absorption bands of the Yb^3+^ and various Sm^3+^ transitions are labelled in Figure 4a,b. About seven of the 4f^5^-4f^5^ electronic transitions of the Sm^3+^ from the ^5^H_5/2_ ground state to different excited states are observed in the hypersensitive frequency, as shown in Figure 4b. The Sm^3+^-hypersensitive absorption bands occur as a result of the transition from the ^5^H_5/2_ ground state to excited states such as ^6^F_11/2_, ^6^F_9/2_, ^6^F_7/2_, ^6^F_5/2_, ^6^F_3/2_, ^6^H_15/2_, and ^6^F_1/2_, which correspond to wavenumbers of 10,333 cm^−1^ (968 nm), 9238 cm^−1^ (1082 nm), 811 cm^−1^ (1233 nm), 7244 cm^−1^ (1381 nm), 6731 cm^−1^ (1486 nm), 6513 cm^−1^ (1536 nm), and 6292 cm^−1^ (1589 nm), respectively. In addition, the intense and broad absorption band in the wavenumber range of 11,200–9601 cm^−1^ are attributed to the overlap transitions between Sm^3+^ (^5^H_5/2_ → ^6^F_11/2_) and Yb^3+^ (^2^F_7/2_ → ^2^F_5/2_). The NIR absorption transitions are sensitive to an environmental change around the ions, thus leading to broad absorption ranging from 7542 to 9560 cm^−1^ as the Yb^3+^ content increases to 1 mol%. The broad absorption band in the SBT4 glass could be accredited to the presence of a -OH second overtone in the glass. In addition, most of the NIR transitions from ^5^H_5/2_ → ^6^F_9/2_, ^6^F_7/2_, ^6^F_5/2_, ^6^F_3/2_, and ^6^F_1/2_ in Figure 4 obey the selection rule with ∆J≤6 inducing electric-dipole interactions, thus exhibiting sharp and more intense absorption bands.

Furthermore, the absorption spectra of the as-prepared glasses were employed to investigate the significant nature of the bonding between Sm^3+^ and their surrounding oxygen atoms in the host glasses. The slight change in the absorption peak position at the NIR transitions of the glasses is attributed to the variation in the surrounding environment of Sm^3+^ and the structure complexity. These were evaluated using the nephelauxetic ratio (β) and bonding parameters (δ). The nephelauxetic effect arises from the expansion of partially filled 4f-4f-shells owing to different glass compositions with a charge transfer from the ligand to the core of the Sm^3+^ and Yb^3+^ [39,40]. The nephelauxetic ratio, (β), is expressed as the ratio of a particular RE^3+^ ion-absorption transition in wavenumber (cm^−1^), $\left(v_{gl}\right)$, in the host glass, to the corresponding transition of the aquo-ion in wavenumber (cm^−1^), $\left(v_{a}\right)$ [5,41,42]:(3)$$\beta =\frac{v_{gl}}{v_{a}}$$

The average value of the nephelauxetic ratio, $\bar{\beta }=\frac{\beta }{N}$, (*N* is the total amount of observed absorption in the glass matrix) was utilised to determine the bonding parameter, (σ), from the following equation [17,41,42]:(4)$$\sigma =\left(\frac{1-\bar{\beta }}{\bar{\beta }}\right)\times 100$$

The σ can be either positive or negative, which signifies the covalent or ionic nature of the RE^3+^-ligand bond depending on the surrounding environment. All absorption coefficient spectra of the glass samples were fitted with multiple Gaussian function to determine the peak positions of each NIR transition of the Sm^3+^ ions. This provides an accurately centred wavenumber for each transition and enables accuracy in assessing the nephelauxetic ratio (β) and bonding parameters. The nephelauxetic effect is caused by the shortening of the metal–ligand distance bond formation due to the expansion of the electron cloud, which leads to a decrease in the coordination number. Table 3 shows β and σ values obtained from Equations (3) and (4) for the SBT glass series. The negative values of σ confirm the ionic character of the Sm^3+^- and Sm^3+^:Yb^3+^-doped and -codoped barium fluorotellurite glasses. The ionicity, σ, values increased with the increasing Yb^3+^ content leading to a structural modification. Such structural alterations have been observed in various glasses, as shown in Figure 2, leading to an increase in the TeO_4_ bridge oxygen peak, which accounts for increasing bonding parameters with increasing Yb^3+^.

### 3.3. Photoluminescence Spectra and Energy Transfer Process

The photoluminescence emission spectra of various Yb^3+^ concentrations in the Sm^3+^-doped barium fluorotellurite glass were initially investigated in the wavelength range 500–1600 nm. Under the 450 nm excitation, the exciting radiation undergoes a ground state absorption from a ^6^H_5/2_ state to a higher-lying ^4^G_7/2_ state with stark levels of Sm^3+^ (acting as a sensitizer). The populated ^4^G_9/2_ level dissipates with no emission light via multiphonon relaxation and a nonradiative process to lower levels such as ^4^G_9/2_ → ^4^G_7/2_, ^4^F_3/2_, and ^4^G_5/2_ shown in Figure 5. The emitted light undergoes radiative decay from the ^4^G_5/2_ energy level to various lower-lying states. On the other hand, the energy absorbed by the Sm^3+^ can also be transferred to an acceptor, Yb^3+^ (^4^G_9/2_ → ^2^F_5/2_ or ^4^F_7/2_ → ^2^F_5/2_), through energy transfer (ET) and cross-relaxation (CR) processes as illustrated in Figure 5. Figure 6 shows photoluminescence emission intensity spectra of the four different Yb^3+^ concentrations in the Sm^3+^-doped fluorotellurite glasses recorded at room temperature under a 450 nm wavelength excitation. An average diode laser power of 60 mW was employed to excite the glass samples.

Figure 6a,b represent the photoluminescence spectra of the Sm^3+^-doped and Sm^3+^:Yb^3+^-codoped barium fluorotellurite glass series in the visible and near-infrared spectral ranges. The visible photoluminescence emission spectra reveal four broad and intense peaks centred at 562 nm (yellow-green band), 598 nm (orange band), 646 nm (red band), and 707 nm (red band), attributing to the 4f-4f interconfigurational transitions. These emission peaks correspond to the ^4^G_5/2_ → ^6^H_5/2_, ^4^G_5/2_ → ^6^H_7/2_, ^4^G_5/2_ → ^6^H_9/2_, and ^4^G_5/2_ → ^6^H_11/2_ transition bands shown in Figure 5. Similarly, the NIR photoluminescence emission spectrum peaks of the Sm^3+^-doped glass centred at 906, 949, 1030, 1130, 1200, 1293, and 1418 nm correlating with the ^4^G_5/2_ state to ^6^F*_J_* (*J* = 3/2, 5/2, 7/2, 9/2) and the ^6^F_11/2_ state to ^6^H*_J_* (*J* = 7/2, 9/2, 11/2) transitions. It is essential to mention that this is the first report on Sm^3+^ photoluminescence emission peaks occurring at 1200, 1293, and 1418 nm. In addition, fluorotellurite glasses codoping with Sm^3+^:Yb^3+^ exhibit a broad photoluminescence emission band range between 850 and 1100 nm owing to the overlap of the Sm^3+^ (^4^G_5/2_ → ^6^F_3/2_, ^6^F_5/2_, ^6^F_7/2_) and Yb^3+^ (^2^F_5/2_ → ^2^F_7/2_) transitions. It can be seen that the photoluminescence emission bands measured at the visible and NIR decrease with an increasing Yb^3+^ concentration, which could be ascribed to concentration quenching. The photoluminescence emission of sample SBT1 was collected without and with a 495 nm short-pass filter in the NIR wavelength using a 976 nm excitation for comparison as shown in Figure 7a. No significant difference is observed between a 495 nm filter-based and without photoluminescence emission measured under the same laser pump power. Eight transitions are observed in the Sm^3+^-doped barium fluorotellurite glass around 888, 1126, 1173, 1243, 1350, 1386, 1461, and 1486 nm, which are assigned to the ^4^G_5/2_ → ^6^F_3/2_, ^4^G_5/2_ → ^6^F_9/2_, ^6^F_9/2_ → ^6^H_7/2_, ^4^G_5/2_ → ^6^F_9/2_, ^4^G_5/2_ → ^6^F_11/2_, ^6^F_11/2_ → ^6^H_11/2_, ^6^F_9/2_ → ^6^H_9/2_ transitions, respectively. The intense broadband emission peak between 900 and 1050 nm is attributed to the 976 nm laser first-order. Nevertheless, these transitions were only detected from the Sm^3+^ single-doped glass, which contradicts the results of the 1.0 mol% Yb^3+^, 0.5 mol% Sm^3+^-codoped glass reported by Bolton et al. [25].

Figure 7b shows the photoluminescence emission spectra in the NIR (950–1250 nm) region recorded at room temperature under the same experimental conditions using a 976 nm excitation and a 495 nm filter cut-off wavelength. The spectral revealed two emission bands from the Yb^3+^ and Sm^3+^ centred at 1040 and 1125 nm. These emission bands are ascribed to the Yb^3+^ (^2^F_5/2_ → ^2^F_7/2_) and Sm^3+^ (^4^G_5/2_ → ^6^F_9/2_) transitions. The full-width–half-maximum (FWHM) values of the as-prepared Sm^3+^:Yb^3+^-codoped glasses in the NIR emission at 1040 nm were obtained by fitting with the Gaussian curve. The FWHM values measured are ~43, ~40, and ~35 nm, respectively, corresponding to the SBT2, SBT3, and SBT4 glasses.

The photoluminescence–decay curves of samples SBT2, SBT3, and SBT4 for the Yb^3+^:^2^F_5/2_ transitions were measured under a 976 nm excitation at room temperature, as depicted in Figure 7b. The decay profiles were analysed by fitting with a single exponential function to estimate the decay lifetime. Figure 7c illustrates the photoluminescence– decay curves fitted with a single exponential function of the Yb^3+^:^2^F_5/2_ level observed at 1040 nm. Table 4 presents the fitted single exponential decay equation of the decay curves and R-square parameters. According to Figure 7c, the measured lifetimes of 244, 153, and 45 μs were obtained for glass samples SBT2, SBT3, and SBT4, respectively. The monotonically decreased lifetime is attributed to nonradiative decay processes such as concentration quenching, multiphonon relaxation, more efficient energy transfer from the Yb^3+^ to the Sm^3+^, and the coupling of the Yb^3+^:Sm^3+^ to the -OH group. It is important to mention that the lifetimes obtained from these samples from the Yb^3+^:^2^F_5/2_ at 1040 nm in the Sm^3+^:Yb^3+^-codoped glasses are far greater than those reported elsewhere [25].

Figure 8 illustrates the schematical energy transfer processes between the Yb^3+^ and Sm^3+^ ions excited by a 976 nm laser diode. Initially, the 976 nm pump is absorbed from the ground states of the Yb^3+^: ^2^F_7/2_ to the excited states of the Yb^3+^:^2^F_5/2_, respectively. This is followed by spontaneous decay or is de-excited radiatively from the excited states of Yb^3+^:^2^F_5/2_ to ^2^F_7/2_ to emit a photoluminescence emission at 1040 nm. The populated photon in the Yb^3+^:^2^F_5/2_ states can act as an indirect pumping scheme to transfer its energy to the neighbouring Sm^3+^ ions through four main routes such as ET_1_ (Yb^3+^:^2^F_5/2_ → Sm^3+^:^4^G_9/2_), ET_2_(Yb^3+^:^2^F_5/2_ → Sm^3+^:^4^G_5/2_), ET_3_ (Yb^3+^:^2^F_5/2_ → Sm^3+^:^4^G_11/2_), and ET_4_ (Yb^3+^:^2^F_5/2_ → Sm^3+^:^4^G_9/2_) via an energy transfer process or excited state absorption, respectively. Considering that if the Sm^3+^:^4^G_9/2_ energy level dominates in the energy transfer mechanism, it undergoes resonant nonradiative and nonresonant phonon-assisted energy transfer to the Sm^3+^:^4^G_5/2_ level and then is returned radiatively from Sm^3+^:^4^G_5/2_ to Sm^3+^:^6^F_9/2_ to generate a photoluminescence emission at 1125 nm. On the other hand, similar energy transfer processes can occur via ET_3_ and ET_4_ by transferring the majority of the ions in the Yb^3+^:^2^F_5/2_ level to the Sm^3+^:^4^G_11/2_ or ^4^G_9/2_ levels, which may decay nonradiatively and radiatively to ^6^H_7/2_ and emit a photoluminescence emission observed at 1125 nm. Furthermore, the schematic energy-level diagram in Figure 8 of the Sm^3+^:Yb^3+^-codoped glass series shows four possible resonant cross-relaxation processes, CR1, CR2, CR3, and CR4, with a change in energy of practically zero (∆E≈0) as reported elsewhere [25]. The following sequence equations represent these multipolar interactions due to the nonradiative decay: CR1, ^6^F_9/2_ → ^6^F_7/2_ and ^6^H_5/2_ → ^6^H_7/2_ (resonant ∆E≈0); CR2, ^6^F_9/2_ → ^6^F_3/2_ and ^6^H_5/2_ → ^6^H_9/2_ (resonant ∆E≈0); CR3, ^6^F_9/2_ → ^6^H_9/2_ and ^6^H_5/2_ → ^6^F_3/2_ (resonant ∆E≈0); and CR4, ^6^F_9/2_ → ^6^H_7/2_ and ^6^H_5/2_ → ^6^F_7/2_ (resonant ∆E≈0).

## 4. Conclusions

The Sm^3+^- and Sm^3+^:Yb^3+^-doped barium fluoride-substituted zinc tellurite glasses were fabricated and characterised for their numerous properties such as density, refractive index, structural, -OH content, and photoluminescence. Adding the appropriate amount of Yb^3+^ ions into the glass network composition leads to a structural reformation, and the -OH content increases significantly. The photoluminescence emission and decay dynamics of the RE^3+^ ion single-doped and codoped barium fluorotellurite glasses were investigated under 450 and 976 nm band excitations. The photoluminescence emission spectra were recorded using a 450 nm laser diode excitation to monitor the glasses’ visible and NIR emission spectra. This measurement revealed four intense emission peaks at the yellow-green (^4^G_5/2_ → ^6^H_5/2_,), orange (^4^G_5/2_ → ^6^H_7/2_), and red (^4^G_5/2_ → ^6^H_9/2_ and ^4^G_5/2_ → ^6^H_11/2_) bands in the visible spectrum. In contrast, seven emission peaks were observed at the NIR region corresponding to the radiative emission from the ^4^G_5/2_ state to ^6^F*_J_* (*J* = 3/2, 5/2, 7/2, 9/2) transitions and the ^6^F_11/2_ state to ^6^H*_J_* (*J* = 7/2, 9/2, 11/2) transitions. It is observed that the photoluminescence emission intensities measured from visible to NIR decrease with increasing Yb^3+^ concentrations. This quenching process is attributed to resonant energy transfer processes from Sm^3+^ to Yb^3+^ due to a large absorption cross-section of Yb^3+^. Two emission peaks were observed from the Sm^3+^:Yb^3+^-codoped glasses at 1040 and 1125 nm, with a 976 nm excitation correlating to the Yb^3+^ (^2^F_5/2_ → ^2^F_7/2_) and Sm^3+^ (^4^G_5/2_ → ^6^F_9/2_) transitions, whereas the photoluminescence lifetime decreases with increasing Yb^3+^ concentrations. These initial photoluminescence property results of the Sm^3+^:Yb^3+^-codoped glasses suggest new NIR windows for developing lasers suitable for medical diagnostics and imaging in anatomical environments.

## Figures and Tables

**Figure 1 materials-15-03314-f001:**
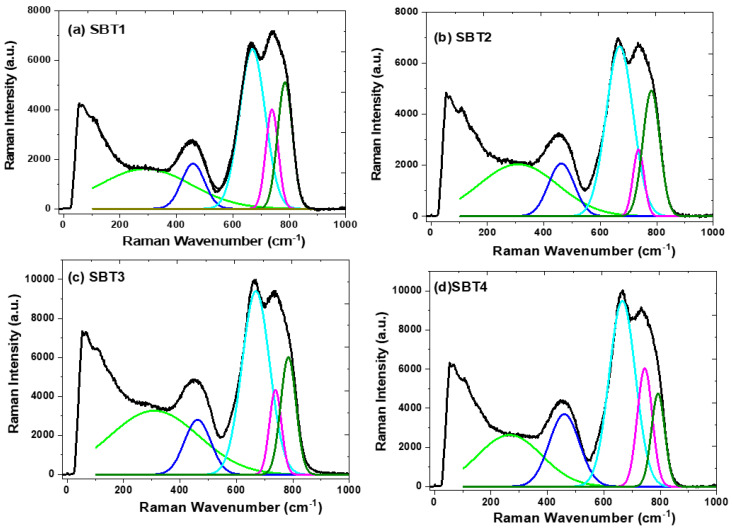
Deconvolution of Raman spectra in four vibration frequency regions of prepared glasses: (**a**) SBT1, (**b**) SBT2, (**c**) SBT3, and (**d**) SBT4.

**Figure 2 materials-15-03314-f002:**
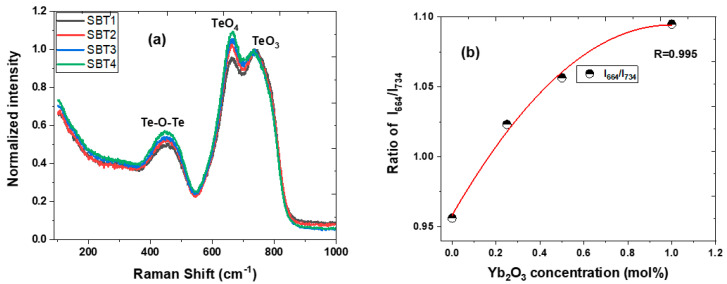
(**a**) Raman spectra of Sm^3+^/Sm^3+^:Yb^3+^-doped barium fluorotellurite glasses, (**b**) corresponding vibration intensity ratio of bridge oxygen to nonbridge oxygen atoms vs. Yb_2_O_3_ concentration.

**Figure 3 materials-15-03314-f003:**
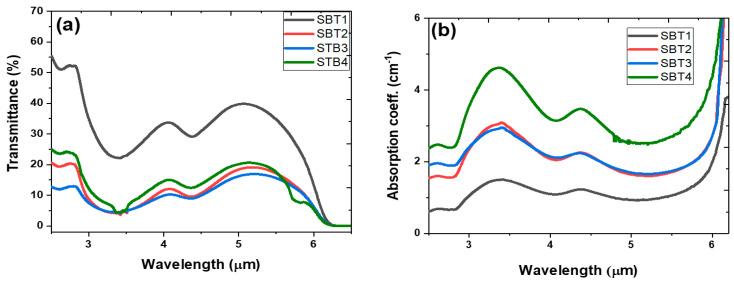
Infrared (**a**) transmission and (**b**) absorption coefficient spectra of Sm^3+^-doped BaF2-TeO2-ZnO glasses by varying Yb^3+^ additive concentration.

**Figure 4 materials-15-03314-f004:**
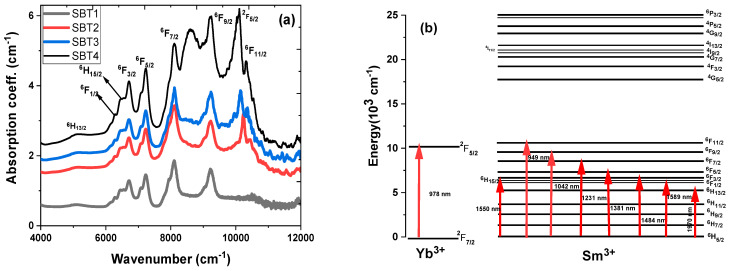
(**a**) Optical absorption coefficient spectra of Sm^3+^ and Sm^3+^:Yb^3+^-codoped barium fluorotellurite glasses and (**b**) corresponding absorption energy-level diagram.

**Figure 5 materials-15-03314-f005:**
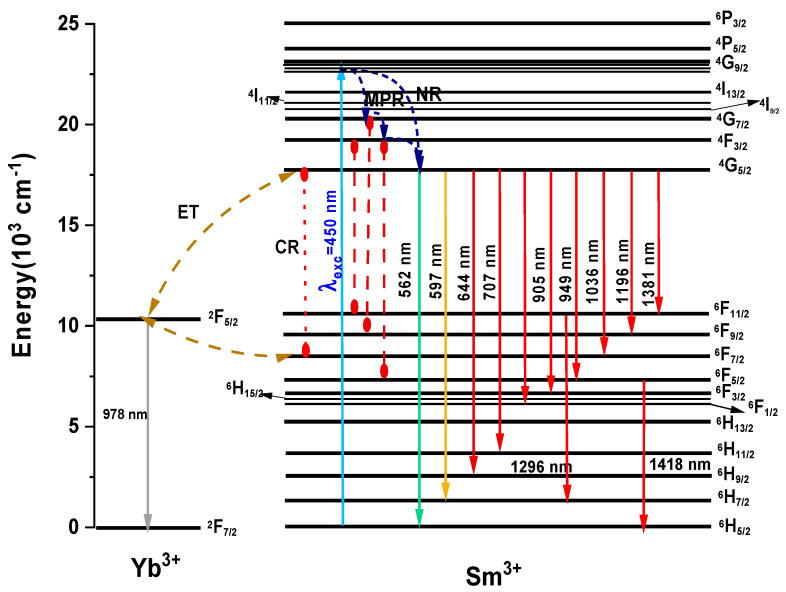
Partial energy-level diagram of Yb^3+^ and Sm^3+^ under 450 nm excitation source showing ground excitation state absorption, nonradiative processes (NR), cross-relaxation (CR), and resonant energy transfer (ET).

**Figure 6 materials-15-03314-f006:**
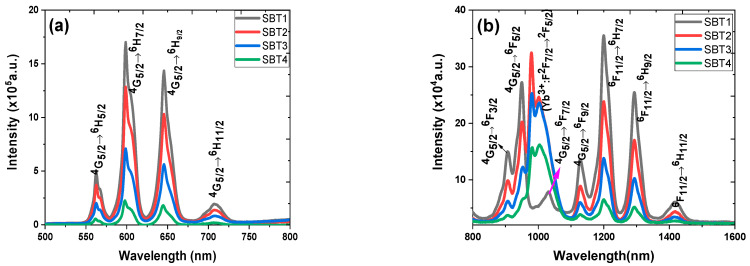
(**a**) Visible and (**b**) near-infrared (NIR) wavelengths photoluminescence spectra of different concentrations of Yb^3+^ in Sm^3+^-doped barium fluorotellurite glass series.

**Figure 7 materials-15-03314-f007:**
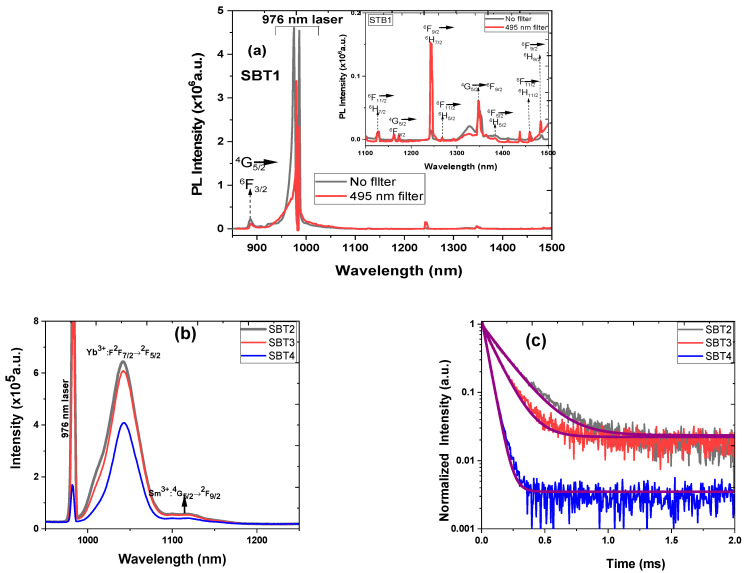
Near-infrared photoluminescence emission spectra of Sm-doped and Sm^3+^: Yb^3+^-codoped glasses under 978 nm excitation source: (**a**) sample SBT1 with 495 short-pass filter and without (an inset shows emission spectra from 1100 to 1500 nm), (**b**) with 450 nm filter, (**c**) photoluminescence–decay curves of different Yb^3+^ content in glasses at ^2^F_5/2_ transition.

**Figure 8 materials-15-03314-f008:**
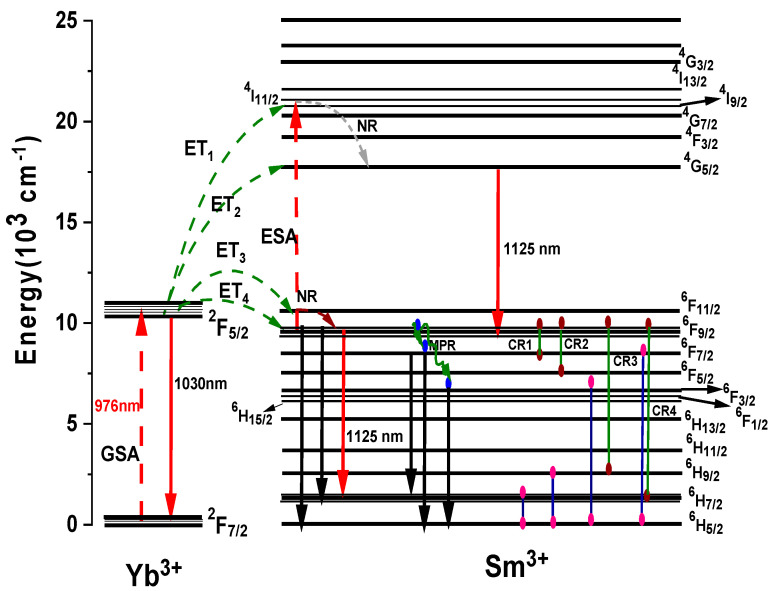
Energy-level diagrams of Sm^3+^ and Yb^3+^ observed in the NIR emission transitions under 976 nm excitation and possible energy transfer processes.

**Table 1 materials-15-03314-t001:** Molar composition, density, and refractive index of the Sm^3+^/Sm^3+^:Yb^3+^-doped barium fluorotellurite glass samples.

Sample ID	Composition (mol%)					Density (g/cm^3^)	Refractive Index
	TeO_2_	ZnO	BaF_2_	Sm_2_O_3_	Yb_2_O_3_		
SBT1	79.50	10	10	0.50	-	5.6221 ± 0.0012	2.0021 ± 0.0005
SBT2	79.25	10	10	0.50	0.25	5.6237 ± 0.0014	2.0060 ± 0.0005
SBT3	79.00	10	10	0.50	0.50	5.6217 ± 0.0037	2.0083 ± 0.0017
SBT4	78.50	10	10	0.50	1.00	5.6183 ± 0.0030	2.0141 ± 0.0013

**Table 2 materials-15-03314-t002:** -OH absorption coefficient, $\alpha_{\mathrm{OH}}$ (cm^−1^), and concentration, $N_{{\mathrm{OH}}^{-}}\left(\mathrm{ions}/{\mathrm{cm}}^{-3}\right)$, at mid-infrared wavelengths of 3.3 and 4.38 μm.

Sample ID	$\alpha_{\mathrm{OH}}\;({\mathrm{cm}}^{-1})$		-OH Concentration (×10^19^ ions/cm^−3^)	
	3.3 μm	4.38 μm	3.3 μm	4.38 μm
SBT1	1.516	1.241	1.859 ± 0.022	1.522 ± 0.018
SBT2	2.973	2.256	3.645 ± 0.041	2.766 ± 0.025
SBT3	3.117	2.278	3.822 ± 0.012	2.793 ± 0.017
SBT4	4.655	3.492	5.707 ± 0.037	4.281 ± 0.006

**Table 3 materials-15-03314-t003:** Transition energy levels and corresponding wavenumber, aquo-ion wavenumber, nephelauxetic effect (β), and bonding parameters.

Transition	Energy Levels (cm^−1^)				Aquo-ion (cm^−1^) [17]
	STB1	SBT2	SBT3	SBT4	
^6^H_5/2_ → ^6^F_11/2_	10,521	10,444	10,354	10,312	10,517
^6^H_5/2_ → ^6^F_9/2_	9240	9229	9244	9240	9136
^6^H_5/2_ → ^6^F_7/2_	8111	8122	8116	8108	7977
^6^H_5/2_ → ^6^F_5/2_	7224	7235	7241	7233	7131
^6^H_5/2_ → ^6^F_3/2_	6719	6719	6719	6714	6641
^6^H_5/2_ → ^6^H_13/2_	6445	6435	6456	6456	6508
^6^H_5/2_ → ^6^F_1/2_	6292	6270	6292	6283	6397
β	7.0238 ± 0.0068	7.0202 ± 0.0046	7.0165 ± 0.0081	7.0078 ± 0.0031	7.0000
$\bar{\beta }$	1.0034 ± 0.0137	1.0029 ± 0.0146	1.0024 ± 0.0150	1.0011 ± 0.0112	1
σ	−0.339 ± 0.002	−0.287 ± 0.005	−0.235 ± 0.008	−0.111 ± 0.004	-

**Table 4 materials-15-03314-t004:** Lifetime decay equation, lifetime, and R-square of the equation.

Sample ID	Decay Equation Profile	Lifetime (μs)	R-Square
SBT2	$I\left(t\right)=0.023+0.969*\exp\left(\frac{t}{0.244}\right)$	244 ± 0.3	0.996
SBT3	$I\left(t\right)=0.022+0.997*\exp\left(\frac{t}{0.154}\right)$	154 ± 0.4	0.996
SBT4	$I\left(t\right)=0.004+1.235*\exp\left(\frac{t}{0.043}\right)$	43 ± 0.1	0.998

## Data Availability

Not applicable.

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
