# Peer review of "Effect of Yb3+ on the Structural and Visible to Near-Infrared Wavelength Photoluminescence Properties in Sm3+-Yb3+-Codoped Barium Fluorotellurite Glasses"

_materials, 2022, doi:10.3390/ma15093314_

Round 1
Reviewer 1 Report
The article by Barimah, Chen, Tenwick, Al-Murish, Sherma and Jha is devoted to composite glasses based on barium fluorotellurite doped with samarium and ytterbium.
The work was done in the classical manner for works devoted to the optical properties of lanthanide-containing glasses. The article is written in good English.
List of remarks:
- Page 2: In addition, Sm3+ has more NIR absorption and photoluminescence emission transitions than erbium (Er3+), neodymium (Nd3+), and ytterbium (Yb3+)
This list of Ln ions is a bit odd. Many lanthanides have IR emissive bands, for example, dysprosium and thulium - but it is strange to isolate samarium due to the number of bands, the possibilities of using IR luminescent derivatives that fall into the transparency windows of telecommunication fibers, biological tissues, etc. are more interesting.
- If I understand correctly, the composition of the glasses after synthesis was not determined, and it was assumed that it corresponded to the inherent stoichiometry. This is not correct, chemical analysis is required (ICP MS, ICP AS, etc.)
- Page 3 Table 1 - it is necessary to give the measurement error for density, Refractive index and chemical composition (after its establishment)
- Page 5, Fig 2a: The peak assigned to TeO3 clearly contains a shoulder on the right. What is it reason of this effect?
- Page 5, Fig 3b: What is the reason for the noise on the green spectrum in the region of 4.5-5 µm?
- Page 6, Table 2: it is required to give the measurement errors of the concentration of OH-groups
- Is the implementation of the Bouguer-Lambert-Beer law in these systems verified (Page 6)?
- Page 8, Table 3: It is required to indicate that the transitions in the table refer to the samarium ion. What is the accuracy of the parameter estimation in the last three rows?
- Fig 5, Page 9: Signatures on the picture (Sm3+ levels, abbreviations MPR, NR) are hard to read, it is necessary to make the picture better and clearer.
- Data labels on Fig.6, Fig.7, Fig. 8 are also unsatisfactory and hard to read
- It is required to bring the fitting curves in figure 7b, the attenuation equation and the parameter R2.
Author Response
Answers to Reviewer’s Comments
Reviewer(s)' Comments to Author
Reviewer: 1 Recommendation: Publish after significant revisions
The article by Barimah, Chen, Tenwick, Al-Murish, Sharma and Jha is devoted to composite glasses based on barium fluorotellurite doped with samarium and ytterbium.
The work was done in a classical manner for works devoted to the optical properties of lanthanide-containing glasses. The article is written in good English.
List of remarks:
- Page 2: In addition, Sm3+ has more NIR absorption and photoluminescence emission transitions than erbium (Er3+), neodymium (Nd3+), and ytterbium (Yb3+)
This list of Ln ions is a bit odd. Many lanthanides have IR emissive bands, for example, dysprosium and thulium - but it is strange to isolate samarium due to the number of bands, the possibilities of using IR luminescent derivatives that fall into the transparency windows of telecommunication fibers, biological tissues, etc. are more interesting.
Ans:
Ans: Although there are several optical transitions in near-IR and mid-IR from a range of RE-ion which are extensively reported, the main point of this article is to identify the gap in the available transition between the edge of the visible and 1300 nm, in which only limited number of rare-earth ions (e.g. Tm, Ho,) have transitions which may be energetically favourable.
The reported optical transitions in Sm3+-ions are unique in this respect which is why we have studied in this article the combination with Yb3+-ion and the impact of optical pumping on near-IR transitions.
1.If I understand correctly, the composition of the glasses after synthesis was not determined, and it was assumed that it corresponded to the inherent stoichiometry. This is not correct, chemical analysis is required (ICP MS, ICP AS, etc.)
Ans: The concentrations of the glasses shown in Table 1 are the corresponding inherent stoichiometry ratio used during glass preparation. After melting the glass, no chemical analysis has been carried out to determine the actual concentration of each constituent oxide and fluoride in the glass. This is because only volatilisation loss occurs through steam and HF gases. We are currently working on further chemical analysis using the XRF and ICP MS technique, which will be a follow-up paper with the Judd-Ofelt parameter analysis of these glasses.
- Page 3 Table 1 - it is necessary to give the measurement error for the values of density, refractive indices and chemical composition (after its establishment)
Ans: The errors in the measurements of density, refractive index, and chemical composition are included in table 1.
- Page 5, Fig 2a: The peak assigned to TeO3 clearly contains a shoulder on the right. What is it reason of this effect?
Ans: ‘The vibrational band centred at 781 cm-1 is ascribed to the stretching mode of nonbridging oxygen TeO3+1 unit’’, which has been added to the manuscript text.
- Page 5, Fig 3b: What is the reason for the noise on the green spectrum in the region of 4.5-5 µm?
Ans: The noise on the green spectrum is due to the FITR instrument changing from one detector to another, which has been corrected in the manuscript.
- Page 6, Table 2: it is required to give the measurement errors of the concentration of OH-groups
Ans: OH-group concentration error analysis has been performed and included in table 2.
- Is the implementation of the Bouguer-Lambert-Beer law in these systems verified (Page 6)?
Ans: Yes, the Lambert-Beer Law in equation (1) of these glass systems is verified using FTIR transmittance or absorbance measurements as shown below. The figures below show the absorption coefficient verse Yb3+ concentration, which exhibits a linear relationship with an R-square parameter >0.93.
- Page 8, Table 3: It is required to indicate that the transitions in the table refer to the samarium ion. What is the accuracy of the parameter estimation in the last three rows?
Ans: As reported in the paper, the slight change in the absorption peak position at the NIR transitions of the glasses is attributed to the variation in the surrounding environment of Sm3+ and structural complexity, which was utilised to estimate nephelauxetic ratio (β) and bonding parameters. From the absorption spectra, the absorption coefficient values for each glass sample were fitted with multiple Gaussian functions to determine the peak positions of each NIR transition of samarium ions. This provides an accurately centred wavenumber for each transition and enables accuracy in assessing nephelauxetic ratio (β) and bonding parameters. Furthermore, error analysis has been carried out and included in table 3.
- Fig 5, Page 9: Signatures on the picture (Sm3+ levels, abbreviations MPR, NR) are hard to read, it is necessary to make the picture better and clearer.
Ans: Figure 5 has been enlarged to make it clear to read.
- Data labels on Fig.6, Fig.7, Fig. 8 are also unsatisfactory and hard to read
Ans: The font size of Figures 6, 7, and 8 have been enlarged to make them easy to read.
- It is required to bring the fitting curves in figure 7b, the attenuation equation and the parameter R2.
Ans: Figure 7 (b) has been fitted with a single exponential function, and the table below shows a single exponential function equation and R-square value.
Sample ID |
Decay equation profile |
Lifetime(μs) |
R-square |
||||||||||||||||
SBT2 Answers to Reviewer’s Comments Reviewer(s)' Comments to Author
The article by Barimah, Chen, Tenwick, Al-Murish, Sharma and Jha is devoted to composite glasses based on barium fluorotellurite doped with samarium and ytterbium. The work was done in a classical manner for works devoted to the optical properties of lanthanide-containing glasses. The article is written in good English. List of remarks:
This list of Ln ions is a bit odd. Many lanthanides have IR emissive bands, for example, dysprosium and thulium - but it is strange to isolate samarium due to the number of bands, the possibilities of using IR luminescent derivatives that fall into the transparency windows of telecommunication fibers, biological tissues, etc. are more interesting. Ans: Ans: Although there are several optical transitions in near-IR and mid-IR from a range of RE-ion which are extensively reported, the main point of this article is to identify the gap in the available transition between the edge of the visible and 1300 nm, in which only limited number of rare-earth ions (e.g. Tm, Ho,) have transitions which may be energetically favourable. The reported optical transitions in Sm3+-ions are unique in this respect which is why we have studied in this article the combination with Yb3+-ion and the impact of optical pumping on near-IR transitions.
1.If I understand correctly, the composition of the glasses after synthesis was not determined, and it was assumed that it corresponded to the inherent stoichiometry. This is not correct, chemical analysis is required (ICP MS, ICP AS, etc.)
Ans: The concentrations of the glasses shown in Table 1 are the corresponding inherent stoichiometry ratio used during glass preparation. After melting the glass, no chemical analysis has been carried out to determine the actual concentration of each constituent oxide and fluoride in the glass. This is because only volatilisation loss occurs through steam and HF gases. We are currently working on further chemical analysis using the XRF and ICP MS technique, which will be a follow-up paper with the Judd-Ofelt parameter analysis of these glasses.
Ans: The errors in the measurements of density, refractive index, and chemical composition are included in table 1.
Ans: ‘The vibrational band centred at 781 cm-1 is ascribed to the stretching mode of nonbridging oxygen TeO3+1 unit’’, which has been added to the manuscript text.
Ans: The noise on the green spectrum is due to the FITR instrument changing from one detector to another, which has been corrected in the manuscript.
Ans: OH-group concentration error analysis has been performed and included in table 2.
Ans: Yes, the Lambert-Beer Law in equation (1) of these glass systems is verified using FTIR transmittance or absorbance measurements as shown below. The figures below show the absorption coefficient verse Yb3+ concentration, which exhibits a linear relationship with an R-square parameter >0.93.
Ans: As reported in the paper, the slight change in the absorption peak position at the NIR transitions of the glasses is attributed to the variation in the surrounding environment of Sm3+ and structural complexity, which was utilised to estimate nephelauxetic ratio (β) and bonding parameters. From the absorption spectra, the absorption coefficient values for each glass sample were fitted with multiple Gaussian functions to determine the peak positions of each NIR transition of samarium ions. This provides an accurately centred wavenumber for each transition and enables accuracy in assessing nephelauxetic ratio (β) and bonding parameters. Furthermore, error analysis has been carried out and included in table 3.
Ans: Figure 5 has been enlarged to make it clear to read.
Ans: The font size of Figures 6, 7, and 8 have been enlarged to make them easy to read.
Ans: Figure 7 (b) has been fitted with a single exponential function, and the table below shows a single exponential function equation and R-square value.
|
244±3 |
0.996 |
|||||||||||||||||
SBT3 |
154±4 |
0.996 |
|||||||||||||||||
SBT4 |
43±0.2 |
0.998 |

Reviewer 2 Report
- Fig (1) in the inset, there is a wrong label (STB4)
- The author wants to prove the existence of the Boson peak in the SBT series by a significant increase in the Raman spectra below 200 cm-1 which is not clear and even if there is a new graph for the four samples together. The author should speak about the Boson peak when we have Raman spectra blow 200 cm-1.
- Page 5: after the figure correct Fig (4) to Fig. (3)
- It is interesting to study the absorption and transmutation in the mid-infrared region and produce ???(cm-1), and concentration, ???-(????/??-3), but what about the range under 2.5 µm. a spectrum from 400 to 4000 cm-1 is required to study a complete infrared spectrum.
- Studying the absorption spectra, and bonding between Sm3+ and surrounding oxygen atoms using FTIR is very useful in the range of more than 800 nm. An absorption spectrum using a UV-Vis spectrophotometer is needed to cover the range from 200 to 2500 nm. There is a very important transition that should take into account in the visible region.
Author Response
Answers to Reviewer’s Comments
Reviewer #2
Comments and Suggestions for Authors
- Fig (1) in the inset, there is a wrong label (STB4)
Ans: The inset in Figure 1 has been labelled correctly.
- The author wants to prove the existence of the Boson peak in the SBT series by a significant increase in the Raman spectra below 200 cm-1 which is not clear and even if there is a new graph for the four samples together. The author should speak about the Boson peak when we have Raman spectra blow 200 cm-1.
Ans: We have repeated the Raman measurements between a spectral range of 0-1000 cm-1 which clearly shows the Boson peak around ~65 cm-1 in all the glass samples prepared. The previous discussion has been modified as indicated below:
‘’ The Boson peak of tellurite glass due to the disordered structure of the amorphous state occurring between 60 and 100 cm-1, is observed in these glass samples. It can be noticed that the Raman spectra of all the glasses prepared exhibit broadband between 0 and 200 cm-1 and a strong peak centred at ~65 cm-1, which confirms the presence of the Boson peak in the SBT series glassy structure. This apparent characteristic peak can be attributed to acoustic phonon and rotational molecular modes [30].’’
- Page 5: after the figure correct Fig (4) to Fig. (3)
- It is interesting to study the absorption and transmutation in the mid-infrared region and produce ???(cm-1), and concentration, ???-(????/??-3), but what about the range under 2.5 µm. a spectrum from 400 to 4000 cm-1 is required to study a complete infrared spectrum.
Ans: The primary purpose of Figure 3 was to evaluate OH- concentrations in the fabricated glasses using the dominant OH- absorption and vibrational bands in the mid-infrared, which dip around 3.3 μm (3000 cm-1) and 4.38 μm (2304 cm-1). Figure 4 illustrates FITR absorption coefficient spectra for all the glasses prepared in the wavenumber spectral range of 4000 to 12000 cm-1, corresponding between 833 nm and 2500 nm.
- Studying the absorption spectra, and bonding between Sm3+ and surrounding oxygen atoms using FTIR is very useful in the range of more than 800 nm. An absorption spectrum using a UV-Vis spectrophotometer is needed to cover the range from 200 to 2500 nm. There is a very important transition that should take into account in the visible region.
- Ans: A UV-Vis-NIR spectrophotometer absorption spectral covers from 250 to 2500 nm wavelength for 0.5 mol% Sm-1.0 mol% Yb codoped tellurite glass has been reported previously by Bolton et al.[26]. We have already measured the absorption spectra of these glasses as shown in the Figure below using a UV-Vis-NIR spectrophotometer. However, we intend to publish this result with the Judd-Ofelt parameter calculation as a follow-up paper, which we are currently working on it.

Round 2
Reviewer 1 Report
The authors revised the article and provided the necessary data. Now the manuscript can be published
Author Response
Not Applicable
Reviewer 2 Report
The manuscript may accept in the present form
Author Response
Not Applicable